# WAVE2 Regulates Actin-Dependent Processes Induced by the B Cell Antigen Receptor and Integrins

**DOI:** 10.3390/cells12232704

**Published:** 2023-11-25

**Authors:** Abhishek Bedi, Kate Choi, Connor Keane, Madison Bolger-Munro, Ashley R. Ambrose, Michael R. Gold

**Affiliations:** Department of Microbiology and Immunology, Life Sciences Institute, University of British Columbia, Vancouver, BC V6T1Z3, Canada; bedi999@mail.ubc.ca (A.B.); madison.bolger-munro@ist.ac.at (M.B.-M.);

**Keywords:** B cell, B cell antigen receptor (BCR), antigen presenting cell (APC), F-actin, WAVE2, Arp2/3 complex, cell spreading, immune synapse

## Abstract

B cell antigen receptor (BCR) signaling induces actin cytoskeleton remodeling by stimulating actin severing, actin polymerization, and the nucleation of branched actin networks via the Arp2/3 complex. This enables B cells to spread on antigen-bearing surfaces in order to increase antigen encounters and to form an immune synapse (IS) when interacting with antigen-presenting cells (APCs). Although the WASp, N-WASp, and WAVE nucleation-promoting factors activate the Arp2/3 complex, the role of WAVE2 in B cells has not been directly assessed. We now show that both WAVE2 and the Arp2/3 complex localize to the peripheral ring of branched F-actin when B cells spread on immobilized anti-Ig antibodies. The siRNA-mediated depletion of WAVE2 reduced and delayed B cell spreading on immobilized anti-Ig, and this was associated with a thinner peripheral F-actin ring and reduced actin retrograde flow compared to control cells. Depleting WAVE2 also impaired integrin-mediated B cell spreading on fibronectin and the LFA-1-induced formation of actomyosin arcs. Actin retrograde flow amplifies BCR signaling at the IS, and we found that depleting WAVE2 reduced microcluster-based BCR signaling and signal amplification at the IS, as well as B cell activation in response to antigen-bearing cells. Hence, WAVE2 contributes to multiple actin-dependent processes in B lymphocytes.

## 1. Introduction

The binding of antigen (Ag) to the B cell antigen receptor (BCR) initiates intracellular signaling that promotes B cell activation, cytokine secretion, B cell proliferation, and differentiation into antibody (Ab)-producing cells [1,2,3,4]. Actin-dependent processes increase the ability of B cells to encounter and respond to Ags [1,5,6,7,8]. Within lymphoid organs, B cells scan the surfaces of follicular dendritic cells and subcapsular sinus macrophages for the presence of their cognate Ag [9,10]. If the B cell encounters its cognate Ag on the surface of these antigen-presenting cells (APC), the resulting BCR signaling initiates actin polymerization and actin network remodeling that allows the B cell to spread and form membrane protrusions that extend across the surface of the APC [5,11]. This enables the B cell to detect more Ag, thereby increasing BCR signaling. BCR-induced cell spreading is enhanced by concomitant integrin signaling, which is especially important when the density of membrane-bound Ag is low [12,13]. When B cells bind Ags that are mobile within a membrane, BCR-Ag microclusters undergo centripetal movement towards the center of the cell–cell contact site and undergo progressive coalescence to form the central supramolecular activation cluster (cSMAC) of an IS [5,11,14]. The centripetal motion and coalescence of BCR microclusters are driven by actin and amplify BCR signaling [6,7,8,11].

The actin remodeling that propels B cell spreading and IS formation involves a large network of proteins [15]. Human loss-of-function mutations in a number of actin-regulatory proteins, including the Wiskott–Aldrich syndrome protein (WASp), the WASp-interacting protein (WIP), the Arpc1B subunit of the Arp2/3 complex, Wdr1, Hem1, and DOCK8 lead to immune dysfunction syndromes that have been termed ‘actinopathies’ [16,17,18,19,20,21]. The loss of these proteins in humans and mice is associated with impaired B cell development, altered BCR signaling, defective B cell activation and Ab responses, and autoimmunity due to defective silencing of self-reactive B cells [22,23,24,25,26,27,28]. Hence, identifying proteins involved in BCR-induced actin dynamics could provide new insights into factors that regulate B cell function in health and disease.

Branched actin networks are nucleated via the Arp2/3 complex, which initiates the formation of new actin filaments that extend at a 70° angle from an existing filament [29,30,31]. Actin polymerization within plasma membrane-proximal branched actin networks exerts outward forces that drive the formation of broad sheet-like membrane protrusions termed lamellipodia [32,33,34]. Due to the elastic resistance of the cell membrane, this is accompanied by centripetal (i.e., retrograde) flow of peripheral actin structures. We have previously shown that Arp2/3 complex activity is essential for B cells to form lamellipodia and undergo radial spreading on anti-Ig-coated coverslips and for B cells to form lamellipodial protrusions when interacting with Ag-bearing cells [11]. Moreover, Arp2/3 complex-mediated actin polymerization generates the actin retrograde flow that drives the centralization of BCR microclusters at the IS [11].

At the plasma membrane, the Arp2/3 complex is activated primarily by WASp family and WAVE family nucleation-promoting factors (NPFs) [35]. The WASp and WAVE NPFs are independently activated by the Cdc42 and Rac GTPases, respectively. Moreover, WASp and WAVE2 have distinct subcellular localizations and functions at the T cell IS [36] and in dendritic cells migrating in 3D environments [37]. The role of WASp in B cell migration, BCR-induced cell spreading, IS formation, and B cell activation has been well documented [38,39,40,41,42,43,44]. Human mutations in WASp, as well as B cell-specific disruption of the gene encoding WASp in mice, lead to B cell-mediated autoimmunity coupled with impaired responses to infections and immunization [39,43,45,46]. Less is known about the role of WAVE proteins in B cells

There are three WAVE proteins, WAVE1, WAVE2, and WAVE3 [47]. RNA-Seq analysis indicates that WAVE2 is the predominant WAVE family protein expressed in murine B cells [48] and human B cells [49]. In leukocytes, WAVE2 is part of the 5-subunit WAVE-regulatory complex (WRC) that consists of WAVE2, Hem1 (Nap1, NCKAP1L), Sra-1, Abi1, and HSCP300 [50,51]. Hem1 and the related Hem2 (NCKAP1) protein are scaffolding proteins that nucleate and stabilize protein complexes. Hem1 is expressed primarily in immune cells whereas other cell types express Hem2 [50]. In mice and humans with loss-of-function mutations in the *NCKAP1L* gene, the resulting Hem1 deficiency leads to aberrant BCR signaling, impaired BCR-stimulated B cell spreading, defective IS formation, and altered Ab responses [25,26,52]. Although these effects could reflect the role of Hem1 in stabilizing the WRC [25,26], Hem1 is also present in other complexes that lack WAVE2 [53,54]. In particular, Hem1 associates with the mTORC2 complex and supports mTORC2-mediated phosphorylation and activation of the Akt kinases [55]. The Hem1-mTORC2-Akt pathway regulates actin polymerization and cell spreading [56,57,58,59]. Because Hem1 could have both WAVE2-dependent and WAVE2-independent functions that impact actin dynamics and organization, we used WAVE2-specific siRNA to directly test the hypothesis that WAVE2 regulates actin-dependent processes in B cells.

## 2. Materials and Methods

### 2.1. B Cells and siRNA Transfection

The A20 IgG^+^ murine B cell line [60] was obtained from ATCC (#TIB-208) and cultured in RPMI-1640 supplemented with 5% heat-inactivated fetal calf serum (FCS), 2 mM glutamine, 1 mM pyruvate, and 50 μM 2-mercaptoethanol (medium). Murine primary B cells were obtained from the spleens of 10- to 13-week-old C57BL/6J mice (Jackson Laboratories, Bar Harbor, ME, USA #000664) using a negative selection B cell isolation kit (Stemcell Technologies, Vancouver, BC, Canada #19854A) as described in [61] and in accord with protocols approved by the University of British Columbia Animal Care Committee. The primary B cells were cultured overnight in a medium containing 5 ng/mL IL-4 (R&D Systems, Minneapolis, MN, USA #404-ML-010) or 5 ng/mL IL-4 plus 5 μg/mL *E. coli* 0111:B4 lipopolysaccharide (LPS; Sigma-Aldrich, St. Louis, MO, USA #L2630) before being used for spreading assays.

A20 cells (1.4 × 10^6^) were transiently transfected with 4 μg of either control non-targeting siRNA (ON-TARGETplus Non-Targeting Pool, Dharmacon, Lafayette, CO, USA #D-00810-01-05) or murine WAVE2/Wasf2 siRNA (SMARTpool ON-TARGETplus, Dharmacon, Lafayette, CO, USA #L-060071-00-0005) using an Amaxa nucleoporator (program L-013) and the Ingenio Electroporation Kit (Mirus Bio, Madison, WI, USA #MIR 50118). The cells were then cultured for 48 h before being used for experiments. Primary B cells were activated with 5 μg/mL LPS for 6 h prior to being transfected with 4 μg of either control siRNA or WAVE2/Wasf2 siRNA using Amaxa nucleoporator program X-001. After transfection, the cells were washed and then cultured for 24 h in medium containing 5 ng/mL IL-4 before being used for experiments. The levels of WAVE2 protein in control siRNA- and WAVE2 siRNA-transfected cells were assessed via immunoblotting (see Section 2.7).

### 2.2. Cell Spreading Analysis

Round 18 mm diameter glass coverslips were coated with PBS containing either 2.4 μg/cm^2^ goat anti-mouse IgG (for A20 cells; Jackson ImmunoResearch, West Grove, PA, USA #115-005-008), 2.4 μg/cm^2^ goat anti-mouse IgM (for primary B cells; Jackson ImmunoResearch, West Grove, PA, USA #115-005-020), 0.15 µg/cm^2^ ICAM-1 (Sino Biological Inc., Beijing, China #50440-M08H), 0.625 μg/cm^2^ goat anti-mouse IgG plus 0.15 µg/cm^2^ ICAM-1, or 2.4 μg/cm^2^ fibronectin (FN; Sigma Aldrich Chemical, St. Louis, MO, USA #F11411) for 30 min at 37 °C. Note that the amount of anti-IgG, anti-IgM, FN, or ICAM-1 that was adsorbed onto the slides was not determined. The coverslips were then blocked for 30 min at room temperature with PBS containing 2% bovine serum albumin (BSA). Primary B cells or A20 cells were resuspended in PBS + 2% FCS (imaging medium) before adding 10^5^ cells (in 100 μL) to each coverslip. Where indicated, A20 cells were pre-treated with 100 µM of the Arp2/3 complex inhibitor CK-666 (Calbiochem, San Diego, CA, USA #CAS 442633-00-3) for 1 h at 37 °C before adding 10^5^ cells (in 100 μL CK-666-containing imaging medium) to the coverslips. At the indicated times, 100 µL of 8% paraformaldehyde (PFA) was added to the coverslips to yield a final concentration of 4%. The cells were fixed with PFA for 15 min and subsequently permeabilized with 0.2% Triton X-100 in PBS for 5 min, both at room temperature. F-actin was visualized by staining with rhodamine-conjugated phalloidin (Thermo Fisher, Waltham, MA, USA #R415, 1:400 in PBS + 2% BSA) for 30 min at room temperature. Coverslips were mounted onto slides using ProLong Diamond anti-fade reagent (Thermo Fisher, Waltham, MA, USA #P36965). Images of the B cell-coverslip interface were captured using a spinning disk confocal microscope (Zeiss Axiovert 200M) with a 100× NA 1.4 oil objective lens. Stimulated emission depletion (STED) microscopy was performed as described previously [62] using a Leica TCS SP8 laser scanning STED system with a 592 nm depletion laser, a CX PL APO 100× NA 1.40 oil objective, and a Leica HyD high sensitivity detector. Images were deconvoluted using Huygens software (version 21.04) (Scientific Volume Imaging, Hilversum, The Netherlands).

The cell area and the percent of the cell area depleted of F-actin were quantified from thresholded binary images using Fiji software (version 2.14.0/1.54f) [63]. The outer edge of the peripheral actin ring was used to define the cell periphery and calculate *A*(*total*), the total cell area. The inner edge of the peripheral actin ring was used to delimit the central actin-depleted region of the cell and calculate its area, *A*(*cleared*). In A20 cells spreading on immobilized anti-IgG, the cell periphery and the inner face of the peripheral actin ring are roughly circular. Hence, the average diameter (D) of the peripheral actin ring could be calculated using the following equation, where R is the average radius from the center of the cell to the outer edge and r is the average radius from the center of the cell to the inner edge of the peripheral actin ring:$$D=R-r=\sqrt{\frac{A(total)}{\pi }}-\sqrt{\frac{A(cleared)}{\pi }}$$

### 2.3. Co-Localization of WAVE2 or the Arp2/3 Complex with F-Actin

A20 cells, as well as primary B cells that had been cultured overnight with either IL-4 or IL-4 plus LPS, were allowed to spread for 30 min on coverslips that had been coated with 2.4 μg/cm^2^ goat anti-mouse IgG (for A20 cells) or 2.4 μg/cm^2^ goat anti-mouse IgM (for primary B cells). The cells were then fixed and permeabilized as described above and blocked with PBS containing 2% BSA for 30 min at room temperature. Cells were stained for 1 h at room temperature with rabbit Abs to WAVE2 (Thermo Fisher, Waltham, MA, USA #PA5-60975, 1:200 in PBS + 2% BSA) or the p34/ARPC2 component of the Arp2/3 complex (Millipore Sigma, Burlington, MA, USA #07-227, 1:200 in PBS + 2% BSA). This was followed by staining with Alexa Fluor 647-conjugated goat anti-rabbit IgG (Invitrogen, Waltham, MA, USA #A-21244, 1:400 in PBS + 2% BSA) plus rhodamine-phalloidin (1:400) for 30 min at room temperature.

### 2.4. Actin Retrograde Flow

A20 cells were co-transfected with a plasmid encoding F-tractin-GFP [64] plus either control or WAVE2 siRNA. After culturing the cells for 48 h, live-cell imaging was performed at 37 °C. Cells (10^5^ in 100 μL imaging medium) were added to coverslips that had been coated with 2.4 μg/cm^2^ goat anti-mouse IgG and allowed to spread for 10 min before imaging the cell-coverslip contact site. Images were acquired every 1 s for 2 min using a spinning disk confocal microscope (Zeiss Axiovert 200M) equipped with a 100× NA 1.40 oil objective lens. Fiji software [63] was used to generate and quantify kymographs.

### 2.5. Visualization of Actomyosin Arcs

A20 cells were transfected with a myosin IIA-GFP plasmid (Addgene, Watertown, MA, USA #38297) [65] and then cultured for 48 h. The cells were added to coverslips coated with either 2.4 µg/cm^2^, 0.625 µg/cm^2^ anti-IgG, 0.15 µg/cm^2^ ICAM-1 or 0.625 µg/cm^2^ anti-IgG plus 0.15 µg/cm^2^ ICAM-1 for 30 min before being fixed with 4% PFA for 10 min and permeabilized with PBS containing 0.2% Triton X-100 for 5 min at room temperature. After staining with rhodamine-phalloidin (1:400) for 30 min at room temperature, the cells were imaged by confocal or STED microscopy. The percent of cells that formed prominent linear actin structures adjacent to the inner face of the peripheral actin ring was determined visually.

### 2.6. BCR Signaling in Response to Immobilized Anti-IgG

Twelve-well tissue culture plates were functionalized with 2.4 μg/cm^2^ goat anti-mouse IgG (Jackson ImmunoResearch, West Grove, PA, USA #115-005-008) and blocked with 2% BSA in PBS as described in Section 2.2. Control siRNA- or WAVE2 siRNA-transfected A20 cells (1.5 × 10^6^ in 120 µL of modified HEPES-buffered saline [mHBS; 25 mM of sodium HEPES, pH 7.2, 125 mM of NaCl, 5 mM of KCl, 1 mM of CaCl_2_, 1 mM of Na_2_HPO_4_, 0.5 mM of MgSO_4_, 1 mg/mL of glucose, 2 mM of glutamine, 1 mM of sodium pyruvate, and 50 µM of 2-mercaptoethanol]) were added to each well of a 12-well tissue culture plate and kept at 37 °C for the indicated times. Cells were lysed by adding 80 µL of RIPA buffer (30 mM of Tris-HCl, pH 7.4, 150 mM of NaCl, 1% Igepal CA-630 (Sigma-Aldrich, St. Louis, MO, USA #I3021), 0.5% sodium deoxycholate, 0.1% SDS, and 2 mM of EDTA) with 2X protease and phosphatase inhibitors (2 mM of phenylmethylsulfonyl fluoride, 20 µg/mL of leupeptin, 2 µg/mL of aprotinin, 2 µg/mL of pepstatin A, 50 mM of β-glycerophosphate, 2 mM of Na_3_MoO_4_, and 2 mM of Na_3_VO_4_). Unstimulated (time 0) cells were kept in suspension for 30 min at 37 °C before being lysed with RIPA buffer and transferring the lysate to an anti-Ig-coated well. All cell lysates were then pipetted vigorously, transferred to Eppendorf tubes, and kept on ice for 10 min with occasional vortexing. After removing insoluble material from the cell lysates via centrifugation, 1/4th volume of 5X SDS-PAGE sample buffer was added to each sample. Signaling protein phosphorylation was assessed via immunoblotting.

### 2.7. Immunoblotting

Cell extracts were separated on 10% SDS-PAGE gels before being transferred to nitrocellulose membranes. The membranes were blocked with 5% milk powder in Tris-buffered saline (10 mM of Tris-HCl, pH 8, and 150 of mM NaCl) and then incubated overnight at 4 °C with rabbit Abs to WAVE2 (Thermo Fisher, Waltham, MA, USA #PA5-60975, 1:200 dilution), phosphorylated CD79a (pCD79a; Cell Signaling Technologies, Danvers, MA, USA #5173, 1:1000), CD79a ([66]; 1:5000), phospho-ERK (pERK; Cell Signaling Technologies, Danvers, MA, USA #9101, 1:1000), ERK (Cell Signaling Technologies, Danvers, MA, USA #9102; 1:1000), phospho-Akt S473 (pAkt; Cell Signaling Technologies, Danvers, MA, USA #9271; 1:1000), Akt (Cell Signaling Technologies, Danvers, MA, USA #9272; 1:1000), or NCKAP1L/Hem1 (Thermo Fisher, Waltham, MA, USA #PA5-58813; 1:500). Immunoreactive bands were visualized using horseradish peroxidase-conjugated goat anti-rabbit IgG (Bio-Rad, Hercules, CA, USA #170-6515; 1:3000) followed by ECL detection (Azure Biosystems, Dublin, CA, USA #AC2010). All Abs were diluted in 5% BSA in Tris-buffered saline and filters were washed with Tris-buffered saline containing 0.1% Tween-20. Blots were imaged and quantified using a Li-Cor C-DiGit imaging system. Full uncropped versions of all blots are shown in Appendix A.

### 2.8. B Cell Responses to Antigen-Bearing Surrogate APCs

Surrogate APCs were generated using protocols described by Wang et al. [61]. Briefly, COS-7 cells (ATCC, #CRL-1651) were transiently transfected using Lipofectamine 3000 (Thermo Fisher, Waltham, MA, USA #L3000008) with a plasmid encoding a single-chain anti-Igκ Ab, as described previously [67]. The single-chain anti-Igκ Ab is a transmembrane protein comprising the single-chain Fv from the 187.1 rat anti-Igκ monoclonal Ab, the hinge and membrane-proximal domains of rat IgG1, and the H-2K^b^ protein transmembrane and cytoplasmic domains [68]. The transfected COS-7 cells were cultured in DMEM supplemented with 5% FCS, 2 mM of glutamine, and 1 mM of pyruvate for 24 h and then plated on FN-coated coverslips overnight so that they could spread and flatten. A20 cells (10^5^ cells in 100 μL of imaging medium) that had been transfected with either control non-targeting siRNA or WAVE2 siRNA were added to COS-7 surrogate APCs expressing the single-chain anti-Igκ as an Ag. After the indicated times at 37 °C, the cells were fixed with 4% PFA for 15 min, permeabilized with 0.2% Triton X-100 in PBS for 5 min, and blocked with PBS containing 2% BSA for 30 min, all at room temperature. The cells were then stained overnight at 4 °C with a rabbit Ab that recognizes the phosphorylated ITAMs in the CD79a subunit of the BCR (anti-pCD79a (Y182); Cell Signaling Technologies, Danvers, MA, USA #5173; 1:400 in PBS + 2% BSA). The coverslips were then incubated for 30 min at room temperature with Alexa Fluor 568-conjugated goat anti-rabbit IgG (Invitrogen, Waltham, MA, USA #A-11036; 1:400) to detect the anti-pCD79a Ab and Alexa Fluor 647-conjugated goat anti-rat IgG (Invitrogen, Waltham, MA, USA #A-21244; 1:400) to detect the rat IgG1 portion of the single chain anti-Igκ surrogate Ag. Coverslips were mounted onto slides using ProLong Diamond anti-fade reagent (Thermo Fisher, Waltham, MA, USA #P36965) and the B cell-COS-7 cell interface was imaged using spinning disk confocal microscopy. A focal plane at the B cell-COS-7 cell interface in which both the Ag cluster fluorescence and the pCD79a cluster fluorescence were maximal was identified visually and then imaged as a single confocal slice of 0.2 µm z-axis thickness. For each B cell, the total fluorescence intensity of clustered pCD79 or clustered Ag at the B cell-COS-7 cell interface was quantified using ImageJ with custom Fiji macros [69], as described previously [70].

APC-induced upregulation of CD69 by primary B cells was assessed as described by Bolger-Munro et al. [11]. Primary B cells were transfected with control siRNA or WAVE2 siRNA as described in Section 2.1 and cultured overnight with IL-4. Alternatively, primary B cells were cultured overnight with IL-4 and then treated with 100 µM CK-666 for 1 h. Anti-Igκ-expressing COS-7 cells (2 × 10^5^ per well) were added to a 12-well tissue culture plate and allowed to adhere for 2 h. The primary B cells (10^6^ per well) were then added to the wells containing the COS-7 surrogate APCs. After co-culturing the B cells and anti-Igκ-expressing COS-7 cells for 18 h, the cells were removed from the wells by pipetting and resuspended in ice-cold FACS buffer (PBS with 2% FCS). Fc receptors on the B cells were blocked by adding 25 µg/mL of the 2.4G2 monoclonal Ab (ATCC, #HB-197) for 5 min. The cells were then stained for 1 h on ice with CD69-PE-Cy7 (eBioscience, #25-0691-82, 1:400) and anti-IgM-FITC (eBioscience, #11-5790-81, 1:400). An LSR II cytometer (Becton Dickinson, Franklin Lakes, NJ, USA) was used to quantify fluorescence. Data were analyzed using FlowJo software (version 10.9.0) (Treestar). Forward and side scatter, as well as IgM-FITC staining, were used to gate on single live B cells. Both the geometric mean fluorescence intensity for CD69 staining and the percent of CD69^+^ cells were determined. The Ag-induced increase in CD69 fluorescence intensity or percent CD69^+^ cells was calculated by subtracting the values for unstimulated B cells that were cultured for 18 h without the COS-7 surrogate APCs.

### 2.9. Statistical Analysis

Prism-GraphPad (version 9.5.0) software was used for statistical analyses. The Mann–Whitney U test was used to compare ranked values in samples with large numbers of cells. Outliers were identified using Robust Regression and Outlier Removal (ROUT) in GraphPad Prism with Q set to 1% [71]. Two-tailed paired *t*-tests and One-Way ANOVA were used to compare values for matched sets of samples, e.g., median values from multiple experiments.

## 3. Results

### 3.1. WAVE2 Regulates BCR-Induced B Cell Spreading

When B cells interact with anti-Ig-coated coverslips, BCR signaling leads to the remodeling of the actin cytoskeleton and the formation of lamellipodia that drive radial cell spreading [11]. This response mimics the initial stages of the B cell-APC interaction and is a robust method for identifying proteins involved in BCR-induced actin remodeling. To investigate the role of WAVE2 in BCR-induced B cell spreading, we used siRNA to deplete WAVE2 in A20 B-lymphoma cells and primary murine splenic B cells (Figure 1A). Compared to control non-targeting siRNA, transfecting the cells with WAVE2-specific siRNA reduced WAVE2 levels by 55–80% (N = 10 experiments) in A20 B-lymphoma cells and by 80–90% (N = 3 experiments) in primary B cells.

A20 cells are widely used to study B cell spreading and BCR-induced actin remodeling. To assess the role of WAVE2 in B cell spreading, control siRNA- and WAVE2 siRNA-transfected A20 were added to anti-IgG-coated coverslips and allowed to spread for 5–30 min, after which the cells were fixed and stained with rhodamine-phalloidin to visualize F-actin. The outer edge of the peripheral actin ring was used to define the cell periphery and quantify the cell area in the lowest confocal plane (i.e., closest to the coverslip). In three independent experiments, the median spreading area of the WAVE2 knockdown (KD) cells at the 15 min time point was significantly lower than that of the control cells (Figure 1B–D). Consistent with the Arp2/3 complex being activated by both the WRC and WASp/N-WASp, the Arp2/3 complex inhibitor CK-666 reduced B cell spreading to a greater extent compared to the depletion of WAVE2 (Figure 1B–D). This suggests that both WAVE2 and WASp/N-WASp contribute to the BCR-induced actin polymerization that drives cell spreading.

Because the depletion of WAVE2 caused a larger reduction in the median spreading area at 15 min than at 30 min (Figure 1D), it suggested that loss of WAVE2 may delay BCR-induced cell spreading. To test this, we compared the spreading areas of control and WAVE2 KD A20 cells over a longer time course. Again, WAVE2 depletion significantly reduced A20 cell spreading at 15 min, but the differences between the control and WAVE2 KD cells became progressively smaller at 30, 45, and 60 min (Figure 1E–G). Thus, WAVE2 contributes mainly to the initial BCR-induced actin polymerization that drives B cell spreading. When WAVE2 is depleted, other Arp2/3 complex activators such as WASp may compensate for the loss of WAVE2 and drive B cell spreading to a similar extent as in control cells, but with slower kinetics.

To confirm that WAVE2 also contributes to BCR-induced spreading in primary B cells, we used siRNA to substantially deplete WAVE2 in primary B cells from mouse spleen (Figure 1A, right panel). When allowed to spread on immobilized anti-IgM for 15 min, WAVE2 KD primary B cells exhibited significantly lower spreading areas than control siRNA-transfected cells (Figure 1H–J). As was observed in A20 cells, the reduction in cell spreading caused by WAVE2 depletion was greater at 15 min than at 30 min, highlighting an important role for WAVE2 in the initial spreading response.

The Hem1 scaffolding protein is an essential component of two different protein complexes that regulate actin dynamics, the WRC [51] and the mTORC2 complex, which regulates actin dynamics via the Akt kinase [52,54,55,56,57]. Because the components of protein complexes often regulate the stability of the other subunits, we asked if the depletion of WAVE2 altered the levels of Hem1. To test this, we transfected A20 cells with the control siRNA or WAVE2 siRNA and then probed the same cell lysates for WAVE2 and Hem1 (Figure 2A). In three independent experiments, we found that Hem1 protein levels in cells transfected with WAVE2 siRNA were 91% ± 14% (mean ± SEM) of those in the control siRNA-transfected cells. Moreover, WAVE2 depletion did not reduce the ability of immobilized anti-IgG to stimulate the phosphorylation of Akt on S473 (Figure 2B), a modification that is mediated by mTORC2. Silencing WAVE2 also had no effect on BCR-induced phosphorylation of the ITAM tyrosine residues in the BCR CD79a subunit or the phosphorylation of ERK (Figure 2B). By analyzing B cell spreading and BCR signaling under the same experimental conditions in which anti-Ig Abs are immobilized on a rigid surface, we have ruled out the possibilities that WAVE2 promotes B cell spreading by globally enhancing BCR signaling or by promoting Akt activation via the Hem1-mTORC2 complex.

### 3.2. WAVE2 Regulates Peripheral Actin Dynamics

Consistent with the idea that WAVE2 promotes actin polymerization at the cell periphery, WAVE2 strongly co-localized with the peripheral actin ring in A20 cells spreading on immobilized anti-IgG (Figure 3A,B). The same was true in primary murine splenic B cells that were allowed to spread on immobilized anti-IgM after being cultured overnight with the survival cytokine IL-4 or with IL-4 plus LPS (Figure 3C,D). LPS-activated primary B cells spread to a greater extent than naïve ex vivo or IL-4-treated primary B cells. We noted that WAVE2 was often enriched in the inner portion of the peripheral actin ring in primary B cells spreading on anti-IgM (Figure 3D).

To visualize how depleting WAVE2 affects the structure of the peripheral actin ring during BCR-induced cell spreading, we utilized STED super-resolution microscopy (Figure 4A). In the WAVE2 KD cells, the peripheral actin filament network appeared to be thinner than in the control cells. Quantitative analysis showed that the average diameter of the peripheral actin ring was significantly lower in WAVE2 KD cells than in control cells after 15 min of spreading on immobilized anti-IgG, with a smaller difference after 30 min of spreading (Figure 4B,C).

Due to the elastic resistance of the plasma membrane, Arp2/3 complex-nucleated actin polymerization at the cell periphery results in a retrograde flow of actin structures towards the center of the cell [11,33]. This can be visualized by live-cell imaging of cells expressing F-tractin-GFP, which dynamically binds to F-actin (Figure 4D). Kymograph analysis, in which the centripetal velocity (Δx/Δt) is calculated for multiple actin tracks, showed that the depletion of WAVE2 reduced the median velocity of the actin retrograde flow (Figure 4E). This is consistent with the loss of WAVE2 resulting in decreased actin polymerization at the cell periphery and decreased cell spreading.

The interaction of WASp and WAVE2 NPFs with the Arp2/3 complex induces a conformational change that enables the Arp2/3 complex to bind to the side of an actin filament and initiate the formation of a new actin branch [72,73]. The decreased spreading and actin retrograde flow in WAVE2 KD cells could, therefore, be due to a reduction in the amount of Arp2/3 complex that is recruited to the cell periphery. To address this, we stained control siRNA- and WAVE2 siRNA-transfected cells for F-actin and the p34/ARPC2 subunit of the Arp2/3 complex. Confocal images revealed substantial overlap of the F-actin and p34 fluorescence signals, with no obvious decrease in the amount of p34/ARPC2 present in the lamellipodial peripheral actin ring of WAVE2 KD A20 cells, as compared to control cells (Figure 5A). To quantify this, we determined the Manders’ coefficient for the fraction of p34/ARPC2 that co-localized with F-actin and found no significant differences between the control and WAVE2 KD cells (Figure 5B). Although the majority of the F-actin in spreading B cells is at the cell periphery, it should be noted that this analysis also includes the co-localization of p34/ARPC2 with F-actin puncta that are present in the central actin-depleted region of the cell. Nevertheless, these findings suggest that WAVE2 regulates peripheral actin polymerization by controlling steps other than the recruitment of Arp2/3 complexes to peripheral actin structures.

### 3.3. WAVE2 Regulates LFA-1-Induced Formation of Actomyosin Arcs

The interaction of B cells with APCs is initiated by the binding of the LFA-1 (α_L_β_2_) and VLA-4 (α_4_β_1_) integrins to ICAM-1 and VCAM-1, respectively, on the APC surface [74]. Subsequent integrin signaling enhances the ability of B cells to spread across Ag-bearing surfaces, allowing the B cell to encounter more Ag, thereby increasing BCR signaling. This is particularly important when the Ag density on the membrane is low [12,13]. To test whether WAVE2 is important for B cell spreading under these conditions, we coated coverslips with ICAM-1 plus a low density (0.625 µg/cm^2^) of anti-IgG Abs. Under these conditions, A20 cell spreading was stimulated by the combination of ICAM-1 + low anti-IgG but not by ICAM-1 alone or by 0.625 µg/cm^2^ anti-IgG alone (Figure 6A).

WAVE2 depletion did not significantly reduce the spreading of A20 cells on ICAM-1 + low anti-IgG at 30 min (Figure 6B), perhaps because LFA-1 can recruit WASp, as shown in NK cells [75]. However, the loss of WAVE2 impaired the formation of linear (i.e., not branched) actin structures adjacent to the inner edge of the peripheral actin ring (Figure 6C–E). These actin arc structures are generated by formin-mediated linear actin polymerization and recruit non-muscle myosin [76]. LFA-1 signaling during B cell spreading greatly increases the formation of these actomyosin arcs, compared to cells spreading on anti-Ig alone [76]. We confirmed this by both confocal and STED microscopy. Figure 6C shows that a transfected myosin IIA-GFP fusion protein was highly enriched at the inner face of the peripheral actin ring in A20 cells spreading on ICAM-1 plus low (0.625 µg/cm^2^) anti-IgG but not in cells spreading on coverslips coated with a high (2.4 µg/cm^2^) density of anti-IgG. STED microscopy showed that linear actin structures were present at the inner face of the peripheral actin ring in 5–35% of A20 cells that had spread for 30 min on ICAM-1 + low anti-IgG (0.625 µg/cm^2^) but were not evident in cells that had spread for 30 min on 2.4 µg/cm^2^ anti-IgG (Figure 6D,E). Importantly, the depletion of WAVE2 significantly reduced the percent of cells that formed actomyosin arcs when they spread on ICAM-1 + low anti-IgG (Figure 6E). This suggests that WAVE2-dependent branched actin polymerization is a prerequisite for the LFA-1-stimulated formation of actomyosin arcs.

### 3.4. WAVE2 Regulates Integrin-Induced B Cell Spreading on Fibronectin

To determine whether WAVE2 also regulates actin dynamics downstream of other receptors in B cells, we investigated whether WAVE2 promotes integrin-mediated B cell spreading and actin remodeling. When B cells bind to the extracellular matrix component FN, signaling by the β1 integrin VLA-4 induces B cell spreading [12,77]. Control siRNA- and WAVE2 siRNA-transfected cells were plated on FN-coated coverslips and then imaged via confocal microscopy (Figure 7A) or STED microscopy (Figure 7B). In contrast to anti-IgG-mediated spreading, A20 cells did not spread radially on FN. Instead, they frequently developed multiple foot-like actin-rich protrusions or assumed an elongated phenotype (Figure 7A,B). These morphological changes were completely dependent on Arp2/3 complex activity (Figure 7A,C,D). When WAVE2 was depleted, the actin networks at the membrane protrusions appeared to be less extensive than in the control cells and the cell spreading area was significantly reduced at the 15 min and 30 min time points compared to the control cells (Figure 7C,D). The control siRNA-transfected cells progressively increased their area between 5 min and 30 min, whereas the WAVE2 KD cells exhibited very little increase in their contact area over that time period (Figure 1C,D). Note that WAVE2 depletion did not reduce the ability of the A20 cells to adhere to FN (Appendix A). Thus, WAVE2 is not essential for VLA-4 activation under these conditions but is important for linking VLA-4-induced signaling to actin remodeling.

### 3.5. WAVE2 Enhances APC-Induced Proximal BCR Signaling and B Cell Activation

When B cells bind APCs that display their cognate Ag, Arp2/3 complex-dependent actin retrograde flow drives the coalescence of BCR-Ag microclusters, which amplifies microcluster-based BCR signaling [11]. Treating B cells with either the Arp2/3 complex inhibitor CK-666 or depleting components of the Arp2/3 complex using siRNA severely impairs the coalescence of BCR microclusters and reduces APC-induced BCR signaling and APC-induced B cell activation [11]. Depleting or inhibiting the Arp2/3 complex also reduces the amount of microcluster-associated proximal BCR signaling, (i.e., phosphorylation of the BCR ITAMs) per unit of Ag that is gathered into clusters [11]. This ratio is a measure of BCR signal amplification. Because WAVE2 is an upstream activator of the Arp2/3 complex, we hypothesized that WAVE2 depletion would reduce APC-induced BCR signaling and BCR signal amplification.

To assess APC-induced BCR signaling, we used a surrogate APC model in which COS-7 cells express a transmembrane form of a single-chain anti-Igκ Ab on their surface [61]. This anti-Igκ Ab acts as a surrogate Ag by binding to the Igκ light chain of the BCR and initiating BCR signaling. A20 cells transfected with either the control or WAVE2 siRNA were added to these COS-7 cells for 5-30 min before fixing the cells and staining for pCD79 and the anti-Igκ surrogate Ag. Images were obtained for a single 0.2 µm-thick confocal slice at the B cell-COS-7 cell interface in which the fluorescence intensity of both the clustered Ag and clustered pCD79 were maximal. In both the control cells and WAVE2 KD cells, Ag and pCD79 clusters formed within 5 min and overlapped extensively (Figure 8A), which is indicative of Ag-bound BCRs with phosphorylated ITAMs. Quantifying the amount of clustered pCD79 at the B cell-COS-7 cell contact site for each B cell showed that WAVE2 KD cells consistently exhibited significantly less microcluster-based BCR signaling than control cells at the 5 min and 30 min time points (Figure 8B,C). Less consistent reductions in BCR signaling were observed at the 10 min and 15 min time points (Figure 8B,C). Similarly, the ratio of clustered pCD79 divided by clustered Ag for each B cell was significantly reduced at the 5 min and 30 min time points when WAVE2 was depleted (Figure 8D,E). Hence, WAVE2 contributes to the actin-dependent BCR signal amplification that occurs when B cells bind to Ags that are mobile within a membrane. In contrast, WAVE2 depletion did not reduce BCR signaling induced by immobilized anti-Ig Abs (Figure 2B). Although both mobile and immobile Ags stimulate actin-retrograde flow, BCR microclusters that are bound to immobilized Ags do not undergo the centripetal movement and progressive coalescence that amplifies BCR signaling when the Ag is mobile [78].

Because the depletion of WAVE2 reduced BCR signaling in response to anti-Igκ-expressing surrogate APCs, we tested whether this translated into reduced B cell activation in response to this membrane-bound Ag. To test this, we transfected primary mouse B cells with the control or WAVE2 siRNA, or treated the cells with CK-666, prior to adding them to anti-Igκ-expressing COS-7 surrogate APCs. We then quantified the expression of the activation marker, CD69, on IgM^+^ B cells after 18 h of co-culture (Figure 9A). The increases in CD69 mean fluorescence intensity and the percent of CD69^+^ cells in the control, WAVE2 KD, and CK-666-treated cells were calculated by subtracting the values for unstimulated B cells that were cultured in the absence of surrogate APCs (Figure 9B). After co-culture with anti-Igκ-expressing COS-7 cells, control siRNA-transfected primary B cells exhibited robust increases in the mean fluorescence intensity of CD69 staining and in the percent of CD69^+^ cells. Importantly, this increase in CD69 expression was significantly reduced when WAVE2 was depleted and when the Arp2/3 complex was inhibited (Figure 9B). Thus, WAVE2 is important for both proximal BCR signaling and B cell activation.

## 4. Discussion

In B cells, the Arp2/3 complex activity is essential for the BCR-induced actin remodeling that drives B cell spreading and IS formation [11]. WAVE2 activates the Arp2/3 complex [51] and is part of the 5-subunit WRC that includes the Hem1 protein [51]. Although human and mouse loss-of-function mutations in Hem1 impair B cell function [25,26,52], the role of WAVE2 in B cell actin dynamics and B cell activation has not been directly addressed. Hem1 also regulates the mTORC2-Akt pathway [54,55,56,57] and could therefore have both WAVE2-independent and WAVE2-dependent functions. We found that WAVE2 depletion did not reduce the levels of Hem1. Hence, in contrast to previous studies that examined the roles of Hem1 in B cells, we were able to definitively identify processes regulated by WAVE2. We found that silencing WAVE2 reproduced some of the reported effects of loss of Hem1 such as impaired B cell spreading on anti-Ig-coated surfaces and reduced BCR signaling in response to Ags that are mobile within a lipid bilayer [25]. However, we also show that WAVE2 is important for actin retrograde flow in spreading B cells, LFA-1-dependent formation of actomyosin arcs at synaptic contact sites, and B cell spreading on FN. Moreover, WAVE2 enhances both BCR signaling and signal amplification when B cells interact with Ag-bearing cells, and this is important for subsequent B cell activation. Because our siRNA transfection approach only reduced WAVE2 levels in A20 cells by ~70%, we may be underestimating the magnitude of these effects. Nevertheless, our work extends the previous findings on the role of WRC components in B cells and provides the first direct analysis of the role of WAVE2 in B cells (Figure 10).

When B cells bind to surface-bound Ags, initial BCR signaling initiates the formation of actin-based membrane protrusions [11] that allow the cells to scan for additional Ag. In response to anti-Ig Abs or Ags that are immobilized on rigid surfaces, BCR signaling leads to the formation of a peripheral actin ring in which Arp2/3 complex-nucleated branched actin polymerization exerts outward forces against the plasma membrane [33], resulting in the formation of lamellipodia that drive radial spreading [11,28]. Consistent with a role for WAVE2 in activating the Arp2/3 complex at the periphery of spreading B cells, we found that WAVE2 was highly enriched at the peripheral actin ring in both A20 cells and primary B cells, as was the Arp2/3 complex. Similarly, in other cell types, WAVE2 is localized to the lamellipodial actin network where actin polymerization drives membrane protrusion [36,37].

We found that the depletion of WAVE2 reduced the initial BCR-induced spreading of A20 cells compared to control cells. This was most evident at 15 min after the addition of B cells to anti-Ig-coated coverslips and correlated with a reduced thickness of the peripheral actin ring. Because the density of lamellipodial actin networks is related to the amount of outward force generated and the rate of membrane protrusion [80], the thinner peripheral actin ring in the WAVE2-depleted A20 cells may underlie the reduced initial spreading. Observing A20 cells spreading for longer times, up to 60 min, revealed that WAVE2-depleted A20 cells eventually spread to the same extent as the control cells. This is likely due to the activation of the Arp2/3 complex by WASp and N-WASp. Liu et al. showed that both WASp and N-WASp are important for Ag-induced spreading of both primary mouse B cells and A20 cells [81]. Our findings indicate that WAVE2-dependent actin polymerization also contributes to B cell spreading.

Outward forces exerted on the cell membrane by branched actin networks are opposed by the membrane’s elastic resistance to deformation, resulting in a centripetal flow of peripheral actin structures. The velocity of this actin retrograde flow is related to the rate of actin polymerization at the plasma membrane and the density of the lamellipodial actin network [82]. We found that depleting WAVE2 reduced the velocity of actin retrograde flow at the periphery of A20 cells spreading on immobilized anti-Ig, consistent with the role of WAVE2 in promoting peripheral actin polymerization that exerts a force on the cell membrane. Although the velocity of actin retrograde flow was reduced by depleting WAVE2, we previously showed that inhibiting the Arp2/3 complex completely ablated actin retrograde flow in spreading B cells [11]. This suggests that both WAVE2 and WASp/ N-WASp contribute to the Arp2/3 complex-dependent actin polymerization that results in actin retrograde flow.

The WASp and WAVE NPFs have multiple roles in promoting Arp2/3 complex-nucleated branched actin polymerization [73]. First, NPFs bind to the Arp2/3 complexes and induce a conformational change that enables the Arp2/3 complex to bind to existing actin filaments. After recruiting the Arp2/3 complex to the filament, the verprolin homology/central/acidic (VCA) domains of two NPF molecules activate the filament-bound Arp2/3 complex and initiate the formation of a new actin branch. In addition, WAVE2 can act in an Arp2/3 complex-independent manner to elongate actin filaments and increase actin filament density within lamellipodial branched actin networks that are established by the Arp2/3 complex [83,84,85]. We found that WAVE2 depletion in A20 cells resulted in reduced actin polymerization within the lamellipodia with no apparent change in the amount of Arp2/3 complex that was recruited to these peripheral actin structures. Similar observations were reported in WAVE2-deficient B16F1 melanoma cells [84]. WASp, along with residual WAVE2 in the A20 KD cells, may be sufficient to recruit Arp2/3 complexes to actin filaments. However, WAVE2 may be a limiting factor for the subsequent activation of filament-bound Arp2/3 complexes or for the elongation of newly formed actin branches within the peripheral actin network. STED imaging of WAVE2 KD A20 cells spreading on immobilized anti-Ig revealed that the peripheral actin ring was thinner than in control cells but still contained some branched actin structures. More detailed quantitative super-resolution microscopy is required to determine whether the density of actin branches within the peripheral actin network, or actin filament lengths is reduced in WAVE2 KD cells. Finally, recent work has shown that WAVE2 can support the formation of lamellipodia in Arp2-null HL60 cells, although these protrusions were smaller and formed more slowly than in parental HL60 cells [86]. This suggests that WAVE2 can promote lamellipodia formation via both Arp2/3 complex-dependent and Arp2/3 compex-independent mechanisms.

At the IS, binding of LFA-1 to ICAM-1 on the APC induces the formation of contractile actomyosin arcs at the inner edge of the peripheral actin ring [76]. These actomyosin arcs, which are generated by formin-mediated linear actin polymerization, contribute to the centralization of BCR and TCR microclusters at the IS [76,87] and are important for BCR-mediated acquisition of Ags from APC membranes [76]. Surprisingly, we found that WAVE2 depletion decreased the formation of actomyosin arcs in A20 cells spreading on low anti-Ig plus ICAM-1. The binding of ICAM-1 to LFA-1 has been shown to activate Rac1, the upstream activator of WAVE2, at least in T cells [88]. However, the mechanism by which WAVE2 links LFA-1 to the formation of actomyosin arcs is not clear. The actomyosin arcs appear to originate from formin-dependent linear actin filaments that are nucleated at the plasma membrane, traverse the peripheral branched actin network, and then run parallel to the inner edge of the peripheral actin ring [76,87]. Their expansion and maintenance is dependent on the formin mDia1, as well as crosslinking by myosin II [76,87], and is limited by the competition between formins and the Arp2/3 complex for a limited pool of actin monomers [76,87,89]. Whether WAVE2 contributes to actomyosin arc formation by elongating formin-nucleated linear actin filaments, or via other Arp2/3 complex-independent mechanisms, remains to be explored.

At the B cell-APC IS, Arp2/3 complex activity amplifies microcluster-based BCR signaling [11]. This is likely due to the progressive BCR microcluster coalescence that is driven by actin retrograde flow. We found that depleting WAVE2 in A20 cells reduced the velocity of actin retrograde flow as well as microcluster-based BCR signaling in response to Ag-bearing cells. This reduction in BCR signaling at the IS was paralleled by a decrease in the amount of BCR signaling per unit of Ag that was gathered in microclusters, a measure of signal amplification. This was especially evident at the 5 min time point when BCR signaling and BCR signal amplification were near maximal in the control cells. Using CK-666 wash-in and wash-out approaches, we previously showed that Arp2/3 complex-dependent processes that occur within the first few minutes of B cell-APC interactions are important for B cell activation events that occur 18-72 h later, including CD69 upregulation [11]. We now show that depleting WAVE2 also reduces the ability of Ag-bearing APCs to increase the expression of the activation marker, CD69, on primary B cells. Taken together, our results are consistent with a role for WAVE2 in promoting Arp2/3 complex-dependent actin dynamics that amplify BCR signaling at the IS and promote B cell activation.

Finally, we showed that WAVE2 mediates FN-induced actin remodeling and cell spreading in B cells. Although the binding of F-actin to integrin-associated proteins such as talin promotes integrin activation and clustering [90], we found that WAVE2 depletion inhibited A20 cell spreading but did not affect their ability to bind to FN. Thus, under the experimental conditions that we used, WAVE2 was not a major factor in the inside-out signaling that promotes integrin activation. Instead, WAVE2 acted downstream of VLA-4 as part of the outside-in integrin signaling pathway that stimulates actin polymerization and remodeling. B cell spreading mediated by the binding of VLA-4 to VCAM-1 or FN supports cell–cell interactions that promote B cell survival and activation as well as the trafficking of normal and malignant B cells [91]. Previously, in an in vitro model of transendothelial migration, we showed that A20 cells adhere to and spread on the FN basement membrane beneath a monolayer of vascular endothelial cells [92]. Our present findings suggest that WAVE2 may contribute to this process.

WAVE2 is an essential protein in mammals. Ablation of the gene encoding WAVE2 in mice results in embryonic lethality [93] and no human mutations have been reported. The loss of Hem1 primarily affects immune cells since other cell types express the related Hem2 protein [50]. Because Hem1 stabilizes the WRC [25], the WAVE2-dependent processes that we identified in this study are also likely to be defective in patients with Hem1 loss-of-function mutations. Conversely, hypomorphic alleles of WAVE2 or other components of the WRC that do not result in lethality could cause similar defects in B cell development and function that have been described for Hem1 mutations [25,52].

## 5. Conclusions

Our study provides new insights into the role of WAVE2 in linking the BCR and integrins to actin-dependent processes that support B cell activation. We show that the siRNA-mediated depletion of WAVE2 in B cells delays BCR-induced cell spreading, reduces Arp2/3 complex-initiated actin retrograde flow at the cell periphery, inhibits LFA-1-dependent formation of actomyosin arcs, reduces BCR signaling and B cell activation in response to Ag-bearing cells, and inhibits integrin-induced cell spreading on FN (Figure 10).

## Figures and Tables

**Figure 1 cells-12-02704-f001:**
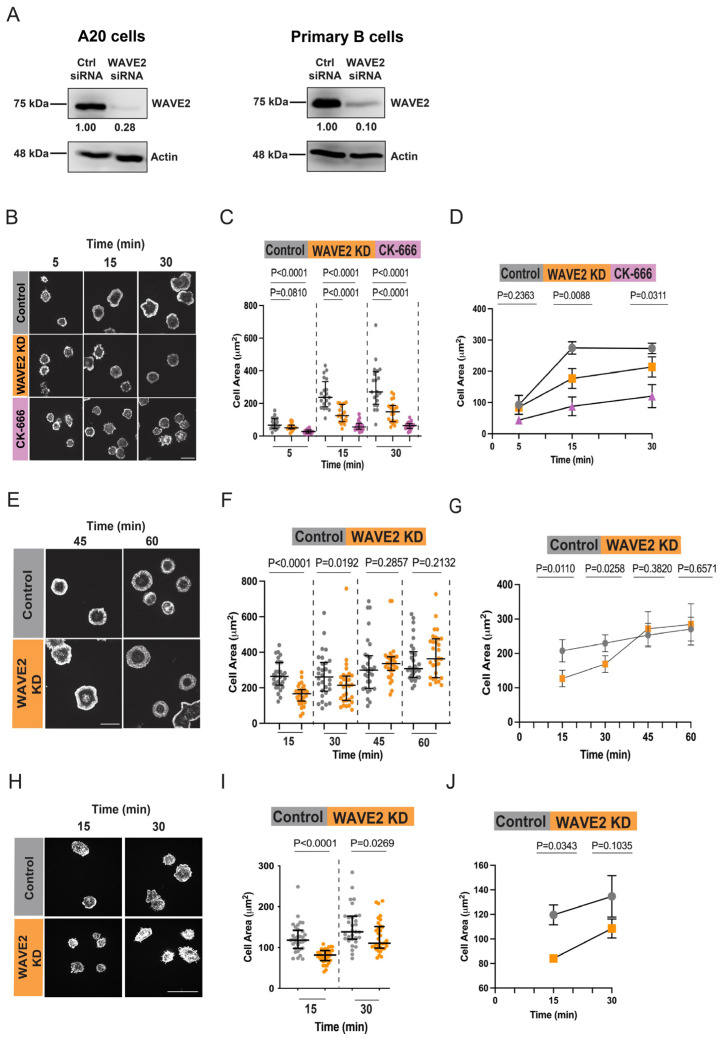
WAVE2 contributes to the ability of B cells to spread on immobilized anti-Ig. (**A**) A20 B-lymphoma cells and primary B cells were transfected with control siRNA or WAVE2 siRNA and analyzed via immunoblotting with Abs to WAVE2 or actin. WAVE2 band intensities were normalized to the actin loading control for the same sample and expressed relative to values for control siRNA-transfected cells. Full uncropped blots are shown in Appendix A. (**B**–**D**) A20 cells were transfected with control siRNA or WAVE2 siRNA (WAVE2 KD) or pre-treated for 1 h with 100 µM CK-666. The cells were added to coverslips coated with 2.4 μg/cm^2^ anti-IgG for the indicated times and then stained with rhodamine-phalloidin to visualize F-actin. Representative images are shown in panel (**B**). Scale bars: 10 µm. Cell areas were quantified using ImageJ version 10.9.0. Panel (**C**) shows one of three independent experiments with similar results. Each dot is one cell, and the medians and interquartile ranges are shown for >25 cells per condition. *p*-values were calculated using the Mann–Whitney U test. Panel D shows combined data from 3 independent experiments. Each symbol is an individual experiment and the data are presented as the mean ± SEM for the median values from the 3 experiments. *p*-values were calculated using two-tailed paired *t*-tests. (**E**–**G**) Control siRNA- and WAVE2 siRNA-transfected A20 cells were added to coverslips that had been coated with 2.4 μg/cm^2^ anti-IgG for the indicated times and then stained with rhodamine-phalloidin. The data are presented as in panels (**B**–**D**). (**H**–**J**) Primary B cells transfected with control siRNA or WAVE2 siRNA (WAVE2 KD) were added to coverslips coated with 2.4 μg/cm^2^ anti-IgM for the indicated times and then stained with rhodamine-phalloidin. The data are presented as in panels (**B**–**D**). All scale bars are 10 µm.

**Figure 2 cells-12-02704-f002:**
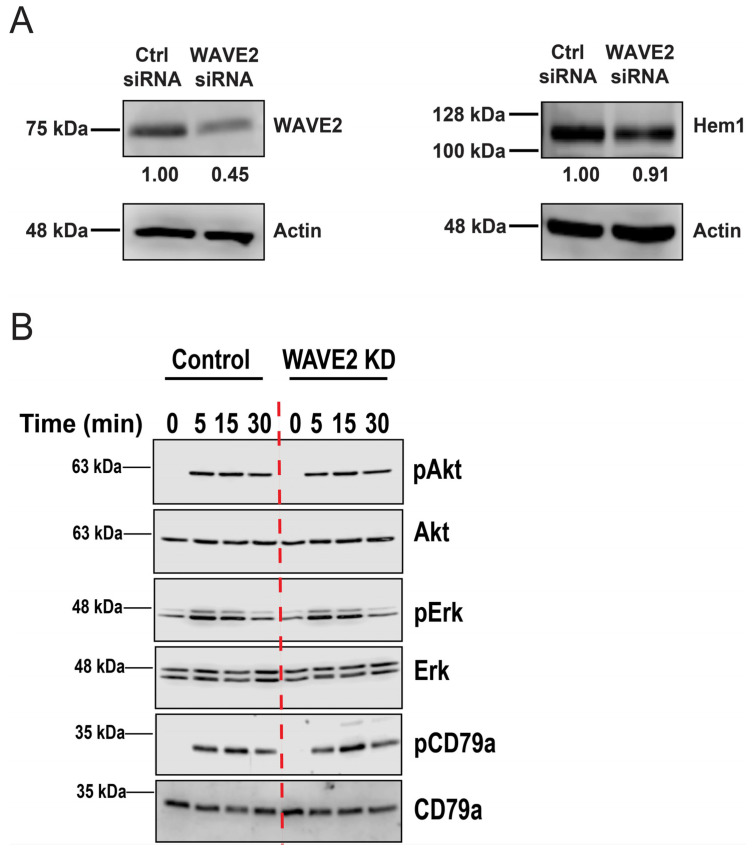
WAVE2 KD does not reduce the expression of Hem1 protein and does not affect BCR signaling in response to immobilized anti-Ig. (**A**) A20 cells were transfected with control siRNA or WAVE2 siRNA, cultured for 48 h, and then analyzed via immunoblotting with Abs to WAVE2, Hem1, or actin (loading control). Normalized levels of WAVE2 or Hem1 relative to control siRNA-transfected cells are shown. (**B**) A20 cells that had been transfected with control siRNA or WAVE2 siRNA were allowed to spread on anti-IgG-coated tissue culture wells for the indicated times before analyzing BCR signaling by immunoblotting for the phosphorylation of the CD79a ITAM (pCD79a) or the phosphorylated (activated) forms of ERK (pERK) or Akt (pAkt). The same cell lysates were probed for total CD79a, ERK, or Akt. The dashed red line was overlaid on the images of the blots to visually separate the time courses for the control siRNA and WAVE2 siRNA-transfected cells. For both panels, one of three independent experiments with similar results is shown. Full uncropped blots are shown in Appendix A.

**Figure 3 cells-12-02704-f003:**
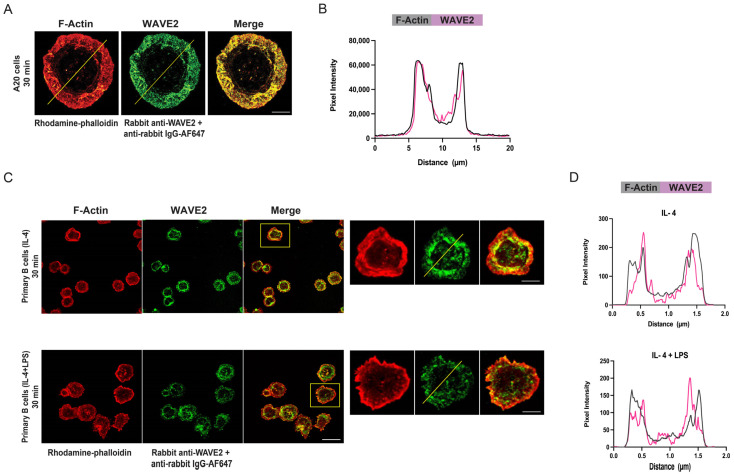
WAVE2 localizes to the peripheral actin ring in spreading B cells. (**A**,**B**) A20 cells were allowed to spread for 30 min on coverslips coated with 2.4 μg/cm^2^ anti-Ig G. The cells were stained for F-actin and WAVE2 and then imaged via STED microscopy (**A**). Scale bar: 5 µm. Panel (**B**) shows F-actin and WAVE2 fluorescence intensity profiles along the yellow lines in panel (**A**). (**C**,**D**) Primary murine B cells that had been cultured overnight with IL-4 or with IL-4 + LPS were allowed to spread for 30 min on coverslips coated with 2.4 μg/cm^2^ anti-IgM. The cells were stained for F-actin and WAVE2 and imaged via confocal microscopy. Enlarged images of the cells indicated by the yellow boxes on the merge panels are shown to the right. Scale bars: 10 µm. Panel (**D**) shows F-actin and WAVE2 fluorescence intensity profiles along the yellow lines in the enlarged cell images in panel (**C**). In the plot profiles in panels (**B**,**D**), the grey lines represent F-actin fluorescence intensity and the purple lines represent WAVE2 fluorescence intensity.

**Figure 4 cells-12-02704-f004:**
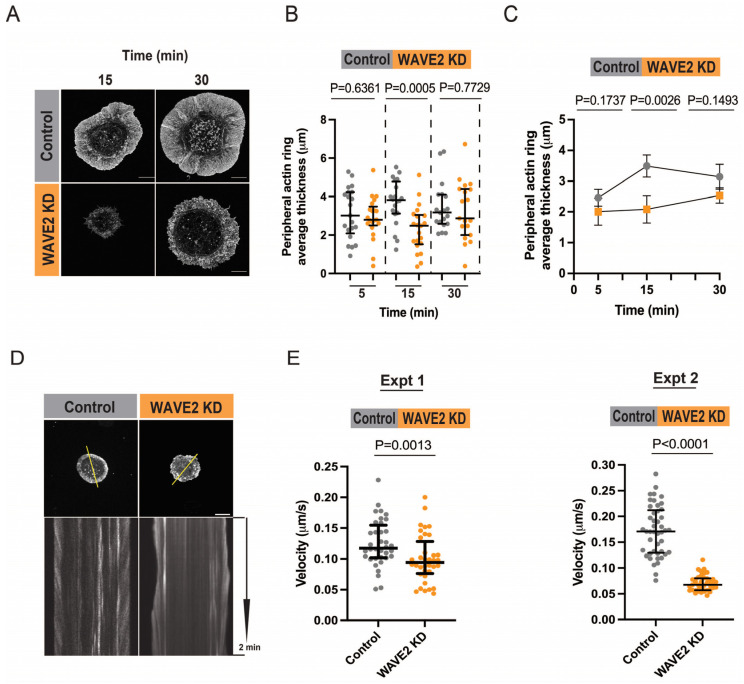
WAVE2 contributes to peripheral F-actin assembly and actin retrograde flow. (**A**) Control siRNA- and WAVE2 siRNA-transfected A20 cells were allowed to spread on anti-IgG-coated coverslips for 15 or 30 min before being stained for F-actin and imaged via STED microscopy. Scale bar: 5 µm. (**B**,**C**) Confocal microscopy images of the control siRNA- and WAVE2 siRNA-transfected A20 cells from the experiments in Figure 1B–D were used to calculate the average peripheral actin ring thickness for each cell, as described in the Methods section. Panel (**B**) shows representative data from a single experiment with the median and interquartile range for >20 cells per condition. *p*-values were calculated using the Mann–Whitney U test. Panel (**C**) shows the mean ± SEM for the median values from 3 independent experiments. *p*-values were calculated using two-tailed paired *t*-tests. (**D**,**E**) A20 cells that had been co-transfected with F-tractin-GFP and either control siRNA or WAVE2 siRNA were allowed to spread for 10 min on coverslips coated with 2.4 μg/cm^2^ anti-IgG before initiating live-cell confocal microscopy imaging. Images were acquired at 1 s intervals for 2 min. In panel (**D**), representative kymographs along the yellow lines in the top panels (Scale bar: 10 µm) are shown in the bottom panels. The centripetal velocity (Δx/Δt) was calculated for individual actin tracks on the kymographs. Panel (**E**) shows two independent experiments in which the actin retrograde flow velocity was calculated for >10 tracks per cell for 3–6 cells. Each dot is one track. Medians and interquartile ranges are shown. The Mann–Whitney U test was used to calculate *p*-values.

**Figure 5 cells-12-02704-f005:**
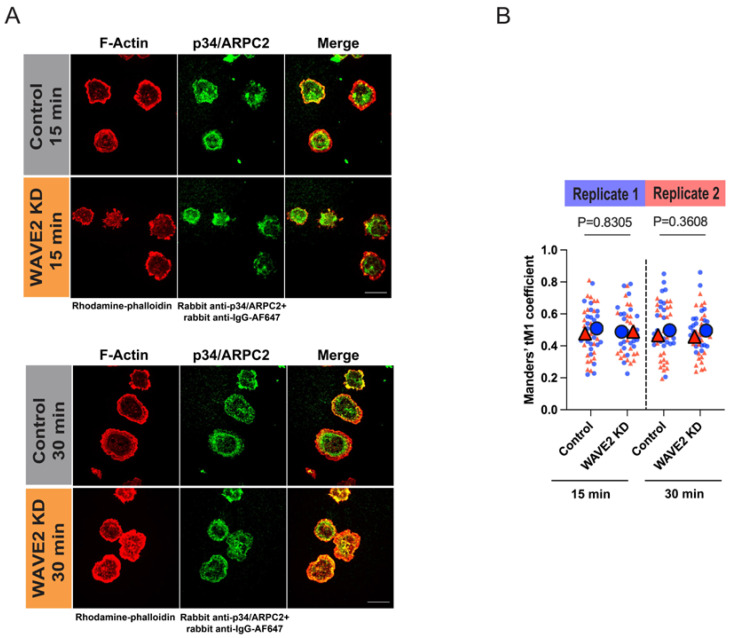
WAVE2 KD does not reduce recruitment of the Arp2/3 complex to actin structures. Control siRNA- and WAVE2 siRNA-transfected A20 cells were allowed to spread on anti-IgG-coated coverslips for 15 or 30 min. The cells were stained for F-actin and the p34/ARPC2 subunit of the Arp2/3 complex and imaged via confocal microscopy (**A**) Representative images. Scale bars: 10 µm. (**B**) The confocal images were used to determine the Manders’ coefficient for the fraction of p34/ARPC2 that co-localized with F-actin. Super-plot showing combined data from 2 independent experiments for >25 cells per condition. Each dot is one cell and the different colors represent the two independent experiments. The large symbols represent the median values for each experiment. *p*-values for control cells versus WAVE2 KD cells in the combined experiments were calculated using the Mann–Whitney U test.

**Figure 6 cells-12-02704-f006:**
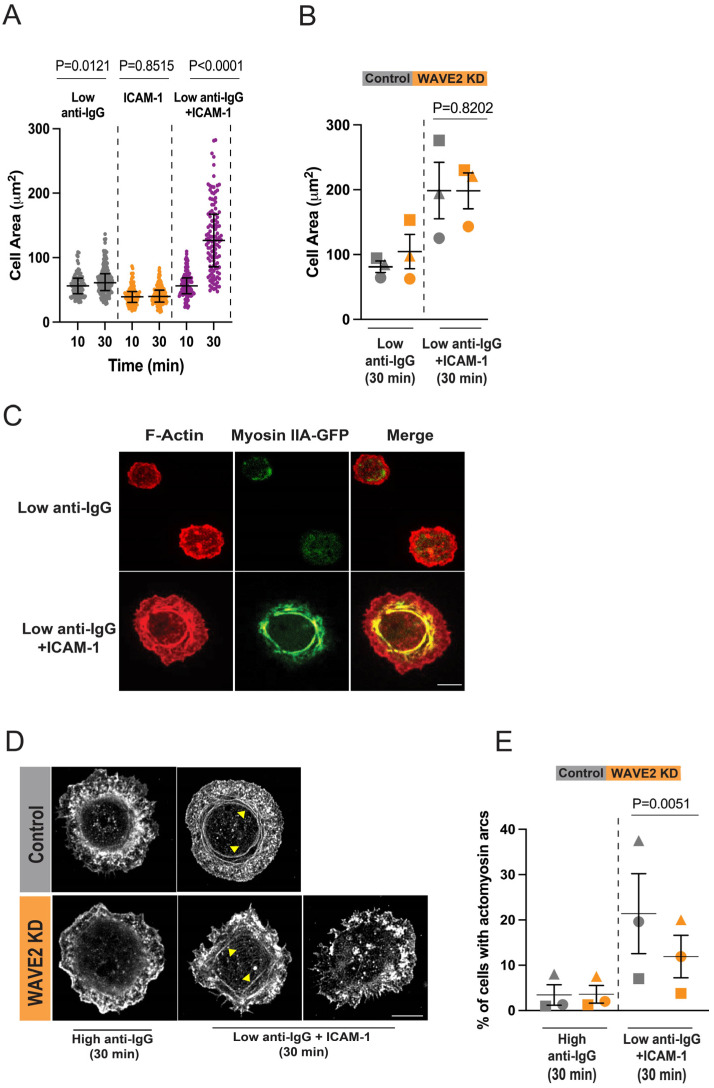
WAVE2 contributes to the LFA-1-dependent formation of actomyosin arcs. (**A**) A20 cells were allowed to spread for 10 or 30 min on coverslips coated with a suboptimal density of anti-IgG (0.625 µg/cm^2^), ICAM-1 (0.15 µg/cm^2^), or 0.625 µg/cm^2^ anti-IgG plus 0.15 µg/cm^2^ ICAM-1. The cells were stained for F-actin and cell areas were quantified from confocal microscopy images. Medians and interquartile ranges are shown for >25 cells per condition. *p*-values were calculated using the Mann–Whitney U test. (**B**) Control and WAVE2 KD A20 cells were allowed to spread for 30 min on coverslips coated with a low density of anti-IgG (0.625 µg/cm^2^) with or without 0.15 µg/cm^2^ ICAM-1. The means ± SEM for the median values from 3 independent experiments are shown. *p*-values were calculated using two-tailed paired *t*-tests. (**C**) A20 cells that had been transfected with a plasmid encoding myosin IIA-GFP were allowed to spread for 30 min on coverslips that had been coated with 2.4 µg/cm^2^ anti-IgG (high anti-IgG) or 0.625 µg/cm^2^ anti-IgG (low anti-IgG) plus 0.15 µg/cm^2^ ICAM-1. Representative images are shown. Scale bar: 10 µm. (**D**,**E**) Control siRNA- and WAVE2 siRNA-transfected A20 cells were allowed to spread for 30 min on coverslips that had been coated with 2.4 µg/cm^2^ anti-IgG (high anti-IgG) or 0.625 µg/cm^2^ anti-IgG (low anti-IgG) plus 0.15 µg/cm^2^ ICAM-1. Cells were stained for F-actin and imaged by STED microscopy. Representative images are shown, and the yellow arrowheads indicate actin arcs (**D**). For comparison, examples are shown of WAVE2 KD cells that did or did not form actin arcs when plated on low anti-IgG plus ICAM-1. Scale bar: 5 µm. In panel (**E**), the percent of cells that formed distinct actin arcs was determined in 3 independent experiments (n > 30 cells per condition). The data are presented as the mean ± SEM for the 3 experiments. *p*-values were calculated using two-tailed paired *t*-tests.

**Figure 7 cells-12-02704-f007:**
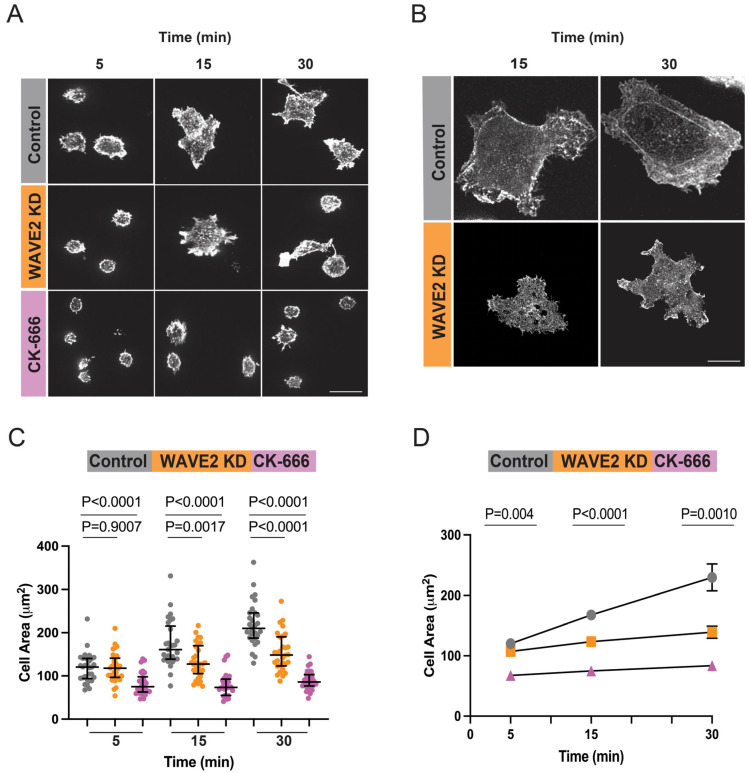
WAVE2 contributes to the ability of B cells to spread on FN. (**A**,**B**) A20 cells that had been transfected with either control siRNA or WAVE2 siRNA, or pre-treated for 1 h with 100 µM CK-666, were added to FN-coated coverslips. After the indicated times the cells were fixed, stained with rhodamine-phalloidin, and imaged via confocal microscopy ((**A**) scale bar: 10 µm) or STED microscopy ((**B**) scale bar: 5 µm). Representative images are shown. (**C**,**D**) In each experiment, cell areas were quantified from confocal microscopy images for >30 cells per condition. A single representative experiment (**C**) as well as combined data from 3 independent experiments (**D**) are shown. The data are presented in as Figure 1. In panel (**D**), *p*-values were calculated for the control cells versus WAVE2 KD cells using two-tailed paired *t*-tests. Where no error bars were visible, they were smaller than the symbols.

**Figure 8 cells-12-02704-f008:**
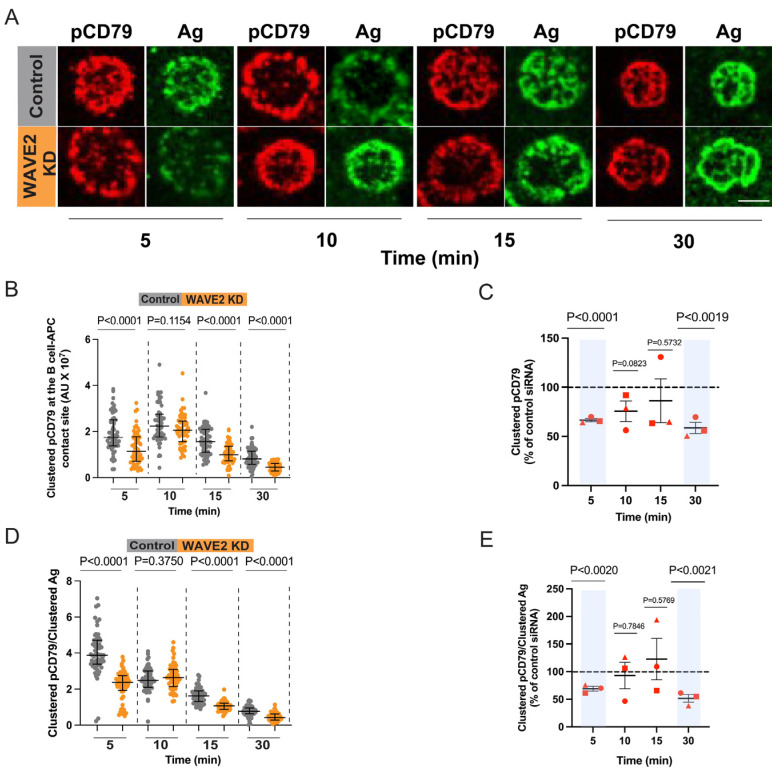
WAVE2 enhances APC-induced BCR signaling and signal amplification. Control siRNA- and WAVE2 siRNA-transfected A20 cells were added to COS-7 cells expressing the ani-Igκ surrogate Ag on their surface. After 5–30 min, the cells were fixed, permeabilized, and stained for pCD79 and the surrogate Ag. (**A**) Representative confocal images of clustered Ag and proximal BCR signaling (pCD79) at the B cell-COS-7 cell contact site. Scale bar: 10 µm. (**B**–**E**) For each B cell, the total amount of clustered pCD79 (panels **B**,**C**) and clustered Ag were quantified for >50 cells per condition and were used to determine the ratio of clustered pCD79/clustered Ag (signal amplification; panels **D**,**E**). Panels (**B**,**D**) show the data from the same representative experiment. Each dot is one cell, and the medians and interquartile ranges are shown. *p*-values were determined using the Mann–Whitney U test. Panels (**C**,**E**) show combined data from 3 independent experiments. Each red symbol is an individual experiment, and the data are presented as the mean ± SEM for the median values from the 3 experiments. *p*-values were calculated using two-tailed paired *t*-tests. In panels (**C**,**E**) the dashed lines represent the values for control cells, which were defined as 100%, and the areas shaded in blue highlight time points for which the values for WAVE2 KD cells were significantly different (*p* < 0.005) from those for the control cells.

**Figure 9 cells-12-02704-f009:**
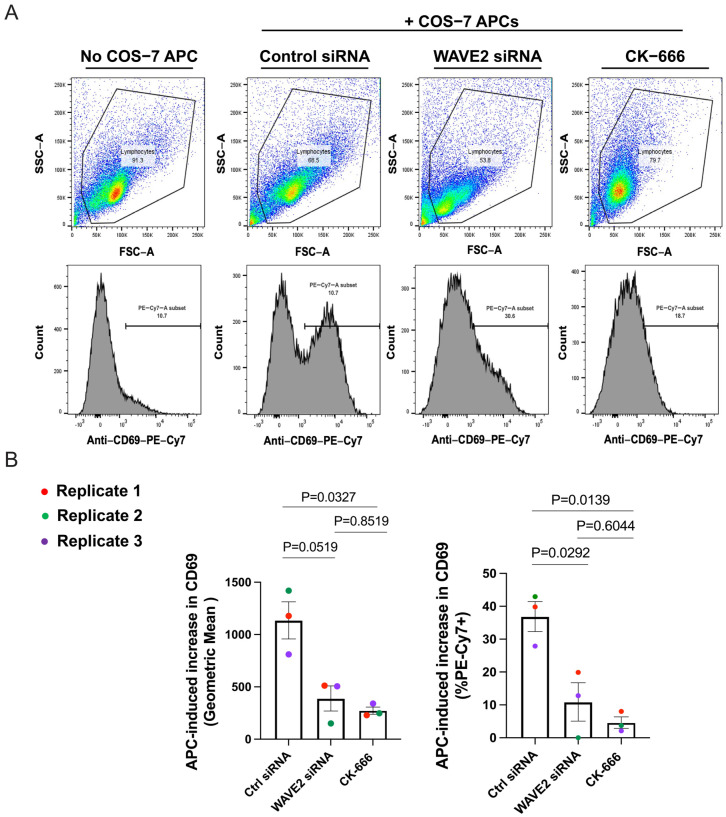
WAVE2 is important for APC-induced B cell activation. Control siRNA- and WAVE2 siRNA-transfected primary B cells, as well as primary B cells that had been treated with CK-666 for 1 h, were added to COS-7 cells expressing the anti-Igκ surrogate Ag. The cells were co-cultured overnight before being stained for CD69 and IgM and analyzed via flow cytometry. (**A**) After gating on IgM^+^ cells, forward/side scatter (top row) was used to identify single live B cells and quantify their CD69 fluorescence (bottom row). (**B**) The surrogate APC-induced increase in cell surface CD69 levels (left panel; geometric means) and percent CD69^+^ cells were calculated by subtracting the values for unstimulated B cells that were cultured without anti-Igκ-expressing COS-7 cells. Each colored dot is an independent experiment. The data are presented as the mean + SEM for the 3 experiments. *p*-values were calculated using the One-Way repeated measures ANOVA test.

**Figure 10 cells-12-02704-f010:**
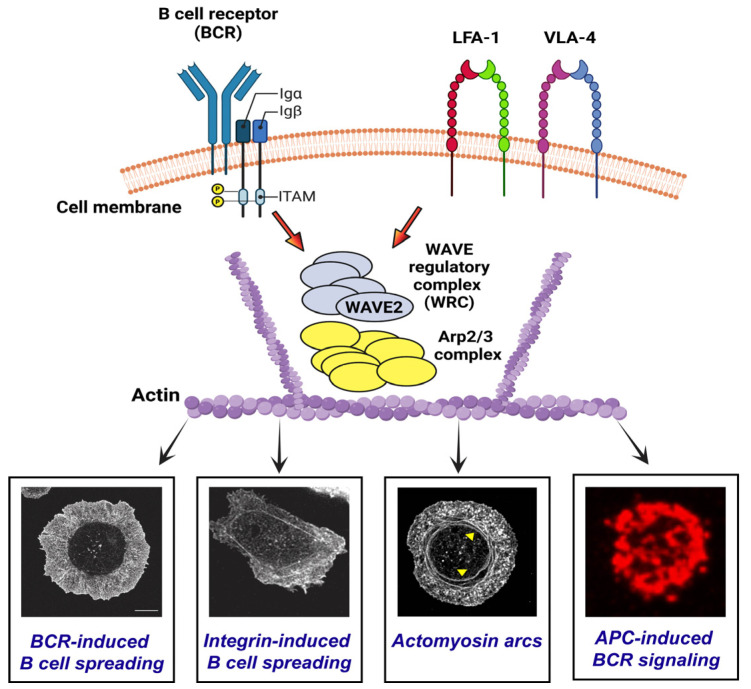
WAVE2 regulates B cell responses. By activating the Arp2/3 complex, the WAVE2-containing WRC contributes to the BCR- and integrin-induced actin remodeling that promotes cell spreading, actomyosin arc formation, and APC-induced BCR signaling. Created with BioRender [79]. The red arrows represent downstream signaling from the BCR and integrins. For the images at the bottom of the figure, the integrin-induced spreading image is taken from Figure 7B, the image showing actomyosin arcs is taken from Figure 6D (the yellow arrowheads point to the actomyosin arcs), and the image for APC-induced BCR signaling is taken from Figure 8A.

## Data Availability

The authors will freely provide all data supporting the published paper upon direct request to the corresponding author.

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
