# Peer review of "WAVE2 Regulates Actin-Dependent Processes Induced by the B Cell Antigen Receptor and Integrins"

_cells, 2023, doi:10.3390/cells12232704_

Round 1
Reviewer 1 Report (Previous Reviewer 2)
Comments and Suggestions for Authors
The revision has addressed all the concerns.
Author Response
We thank the reviewer for their very helpful comments and suggestions, which greatly improved the manuscript.
Reviewer 2 Report (New Reviewer)
Comments and Suggestions for Authors
This manuscript is interesting and solid in its execution. The authors have carried out extensive revisions to address the reviewers comments in a rigorous manner and have included substantial amount of new data. I recommend the manuscript to be accepted for publication.
Author Response
We thank the reviewer for their very helpful comments and suggestions, which greatly improved the manuscript.
This manuscript is a resubmission of an earlier submission. The following is a list of the peer review reports and author responses from that submission.
Round 1
Reviewer 1 Report
Comments and Suggestions for Authors
In this manuscript, the authors addressed the role of WAVE-2, component of the WRC complex, in actin reorganization events triggered by BCR signaling and the integrins LFA-1 and VLA-4 in B lymphocytes. They used the mouse B cell line A20 as experimental model to perform the study, knocking-down WAVE-2 protein expression up to 70% by siRNA techniques at 48 h post-transfection. They compared control and WAVE-2-silenced A20 B cells in assays of immune synapse formation and adhesion to fibronectin by confocal and STED microscopy. The authors found that WAVE-2 is needed for the appropriate assembly and functioning of the F-actin-enriched peripheral structure at the immune synapse, and thus for BCR/antigen-clusters early signaling, estimated by CD79 phosphorylation; WAVE-2 is also involved in the formation of actomyosin arcs at the synapse and for integrin-mediated spreading. The shown data are of high quality, as well as the quantification analysis, and support the conclusions achieved by the authors in the manuscript. Methodology is well-explained and in general the manuscript is well-written.
The novelty of the manuscript resides in the lack of WAVE-2 studies in B cells. It is an important promoter of Arp2/3 complex activity downstream Rac-GTPases, but also recently reported as a negative regulator of mTOR activity in T lymphocytes (Liu et al., 2021, Science, 371: eaaz4544). Loss-of-function mutations in Hem-1, component of the WRC complex and also involved in mTORC2 activity, cause dramatic defects on BCR signaling and B cell immune response, as mentioned by the authors in the manuscript. It is then predictable that WAVE-2 is crucial for B cell homeostasis and function as well. While coming studies with B cell-conditional mouse models will help to address those questions in deepness, this manuscript starts to shed light on the relevance of WAVE-2 for actin rearrangements downstream BCR and integrins in B cells. I have nonetheless some comments to be addressed by the authors.
1. In thymocytes, the lack of Hem-1 provokes the loss of protein expression for the other WRC members WAVE-2, Abi1/2 and Sra-1 (ref. 26 of the manuscript, JEM 2008). What about in A20 B cells when WAVE-2 is knock-down??...are the protein levels of Hem-1, Abi-1/2 and Sra-1 affected somehow? The authors can address that by western blot.
2. The data showed in Figures 1, 3A-B, and 4 correspond to the values of the spreading area at the times indicated. What about frequencies of contacts? It is the frequency of synapses formed affected? …the authors can compare number of cells per field between control and silenced-A20 cells in the acquired images, if they initially have plated the same number of cells. Also, mixed them in 1:1 ratio, plate them on the coverslips, and after the procedures, see if the ratio has been kept or not. This quantification needs to be done in the distinct conditions of BCR stimulation used (different Ab-coating concentrations, plus/minus ICAM-1), and also for FN-coated surfaces. If WAVE-1 knock-down reduces BCR signaling, that might also affect the ability of the cell to establish even a contact with the substrate, and/or the strength of that contact with the substrate.
3. In Figure 5, the data showed a reduction in clustered, phosphorylated CD79 levels at the B cell contact with the APC for silenced-A20 compared to controls. How that translates to downstream signaling events such as ERK/Akt activation, NF-kB, NFAT, and/or FOXO pathways? Are they reduced, enhanced or unaffected by the knock-down of WAVE-2?... having cited above the study on WAVE-2/mTOR/T cells, it will be interesting to measure mTOR activity in BCR-stimulated control versus silenced-A20. The authors can used antigen + ICAM-1 coated beads as APCs instead of COS-7 cells for these experiments.
Reviewer 2 Report
Comments and Suggestions for Authors
This study examined the effects of WAVE2 knockdown by siRNA on the spreading of mouse lymphoma B cell line A20 on antibody-coated in the presence or absence of adhesion molecules and fibronectin-coated glass and BCR phosphorylation induced by cells expressing anti-kappa antibody. The results show that WAVE2 contributes to all these processes probably by facilitating the generation of spreading lamellipodial actin networks, retrograde actin flow, and myosin arcs. WAVE2 is one of the actin nucleation-promoting factors upstream of Arp2/3. While the roles of WASP and N-WASP in B cell activation have been expensively studied, the role of WAVE remains elusive. This manuscript provides information to fill this knowledge gap. Therefore, the manuscript is of interest to the readership of the journal. The manuscript can be further improved by going beyond describing the phenotypes of WAVE2 knockdown under different activation conditions to build mechanistic connections between phenotypes.
Specific comments:
1. This manuscript only used one mouse lymphoma B cell line A20 as the model, which may not completely reflect B cell behavior in vivo. While it is challenging to do WAVE2 knockdown using primary B cells, the expression and distribution of WAVE2 in primary mouse B cells during spreading should be compared with A20 B cells to show whether A20 cells can be an effective model for the study.
2. Does WAVE2 knockdown reduce the recruitment of Arp2/3 to the lamellipodial actin network?
3. The effects of WAVE2 knockdown on B cell spreading, lamellipodial actin networks, actin retrograde flow, and actomyosin arcs were all analyzed using antibody-coated class, but BCR clustering and phosphorylation were done using anti-kappa antibody presented on the cell surface. BCR clustering and phosphorylation should also be analyzed using A20 cells activated by the coated glass to link the changes in actin dynamics with BCR clustering and signaling.
4. WAVE2 knockdown has a much more dramatic impact on B cell adherence to fibronectin-coated glass than B cell spreading on antibody-coated glass. This implies that WAVE2 may have a critical role in B cell migration. However, this is not the focus of the manuscript and thus should be excluded from this manuscript, particularly when there was no following up.
5. How were the antibody, adhesion molecule, and fibronectin coating densities measured? Please include this in the method section.
6. How the amount of clustered antigen and pCD79a at contact sites were measured? How were the imaging planes chosen, as the imaging focal plane is critical? Images in Figure 5A show different sizes of B cell contact areas and different distributions of BCRs under the same condition.
7. Since WAVE2 siRNA only reduced WAVE2 expression, WAVE2 deletion, repeatedly used in the manuscript, is not appropriate. Instead, WAVE2 knockdown should be used.
8. Images in Figure 3C are distorted.